# Particle foraging strategies promote microbial diversity in marine environments

Ali Ebrahimi[1†], Akshit Goyal[2†], Otto X Cordero[1*]

[1]Ralph M. Parsons Laboratory for Environmental Science and Engineering, Department of Civil and Environmental Engineering, Massachusetts Institute of Technology, Cambridge, United States; [2]Physics of Living Systems, Department of Physics, Massachusetts Institute of Technology, Cambridge, United States

**Abstract** Microbial foraging in patchy environments, where resources are fragmented into particles or pockets embedded in a large matrix, plays a key role in natural environments. In the oceans and freshwater systems, particle-associated bacteria can interact with particle surfaces in different ways: some colonize only during short transients, while others form long-lived, stable colonies. We do not yet understand the ecological mechanisms by which both short- and long-term colonizers can coexist. Here, we address this problem with a mathematical model that explains how marine populations with different detachment rates from particles can stably coexist. In our model, populations grow only while on particles, but also face the increased risk of mortality by predation and sinking. Key to coexistence is the idea that detachment from particles modulates both net growth and mortality, but in opposite directions, creating a trade-off between them. While slow-detaching populations show the highest growth return (i.e., produce more net offspring), they are more susceptible to suffer higher rates of mortality than fast-detaching populations. Surprisingly, fluctuating environments, manifesting as blooms of particles (favoring growth) and predators (favoring mortality) significantly expand the likelihood that populations with different detachment rates can coexist. Our study shows how the spatial ecology of microbes in the ocean can lead to a predictable diversification of foraging strategies and the coexistence of multiple taxa on a single growth-limiting resource.

## Editor's evaluation

This manuscript tackles an under-explored area in understanding microbial coexistence in marine and aquatic environments. This manuscript adds to the recently renewed interest on applications of optimal foraging theory to the study of microbial growth on marine snow.

## Introduction

Microbes in nature are remarkably diverse, with thousands of species coexisting in any few milliliters of seawater or grains of soils (*Azam and Malfatti, 2007*; *Young and Crawford, 2004*). This extreme diversity is puzzling since it conflicts with classic ecological predictions. This puzzle has classically been termed 'the paradox of the plankton', referring to the discrepancy between the measured diversity of planktons in the ocean, and the diversity expected based on the number of limiting nutrients (*Ghilarov, 1984*; *Shoresh et al., 2008*; *Hutchinson, 1961*; *Goyal and Maslov, 2018*). Decades of work have helped, in part, to provide solutions for this paradox in the context of free-living (i.e., planktonic) microbes in the ocean. Many have suggested new sources of diversity, such as spatio-temporal variability, microbial interactions, and grazing (*Rodriguez-Valera et al., 2009*; *Muscarella*

*For correspondence:
ottox@mit.edu

†These authors contributed equally to this work

Competing interest: The authors declare that no competing interests exist.

*et al., 2019*; *Saleem et al., 2013*). However, in contrast with free-living microbes, the diversity of particle-associated microbes — often an order of magnitude greater than free-living ones — has been overlooked (*Milici et al., 2017*; *Ganesh et al., 2014*; *Crespo et al., 2013*). In contrast with planktonic bacteria, which float freely in the ocean and consume nutrients from dissolved organic matter, particle-associated microbes grow by attaching to and consuming small fragments of particulate organic matter (POM) (of the order of micrometers to millimeters). It is thus instructive to ask: what factors contribute to the observed diversity of particle-associated microbes, and how do these factors collectively influence the coexistence of particle-associated microbes?

The dispersal strategies of particle-associated microbes can be effectively condensed into one parameter: the rate at which they detach from particles. This rate, which is the inverse of the average time that microbes spend on a particle, is the key trait distinguishing particle-associated microbial populations from planktonic ones *Yawata et al., 2020*; *Fernandez et al., 2019*. The detachment rates of such particle-associated taxa can be quite variable (*Grossart et al., 2003*; *Yawata et al., 2014*). Bacteria with low detachment rates form biofilms on particles for efficient exploitation of the resources locally, while others with high detachment rates frequently attach and detach across many different particles to access new resources (*Ebrahimi et al., 2019*). Therefore, to understand how diversity is maintained in particle-associated bacteria we must be able to explain how bacteria with different dispersal rates can coexist. In this study, we address this question. Specifically, we ask how two populations with different dispersal strategies can coexist while competing for the same set of particles, under a range of conditions relevant for marine microbes.

We hypothesize that dispersal is key to the coexistence of particle-associated microbes and thus might explain their high diversity. The degree of species coexistence on particles depends on the balance between growth and mortality. On particles, net mortality rates can be higher than for planktonic cells because of the large congregation of cells on particles, which exposes them to the possibility of a large and sudden local population collapse. The collapse of a particle-attached population can be induced by a variety of mechanisms, including particles sinking below a habitable zone (*Boeuf et al., 2019*), or predation of whole bacterial colonies by viruses or grazers. For instance, after a lytic phage bursts out of a few cells on a particle, virions can rapidly engulf the entire bacterial population, leading to its local demise (*Ganesh et al., 2014*; *Dupont et al., 2015*; *López-Pérez et al., 2016*). Such particle-wide mortality may kill more than 30% of particle-associated populations in the ocean (*Proctor and Fuhrman, 1991*; *Weinbauer et al., 2009*). The longer a population stays on a particle, the higher the chance it will be wiped out. This trade-off between growth and risk of mortality suggests that there could be an optimal residence time on particles. It is however unclear whether sucha trade-off could enable the coexistence of populations with different dispersal strategies and, if so, under what conditions.

Here, we study this trade-off using mathematical models and stochastic simulations. These models reveal that the trade-off between growth and survival against predation can indeed lead to the stable coexistence of particle-associated microbial populations with different dispersal strategies (in our work, detachment rates). We also study how environmental parameters, such as the supply rate of new particles, determine the dominant dispersal strategy and the range of stable coexistence. Our results show that in bloom conditions, when the particle supply is high, fast dispersers that rapidly hop between particles are favored. In contrast, under oligotrophic conditions, when particles are rare, rarely detaching bacteria have a competitive advantage. Overall, our work shows that differences in dispersal strategies alone can enable the coexistence of particle-associated marine bacteria, in part explaining their impressive natural diversity.

## Results

### Overview of the model

To understand how differences in dispersal strategies affect bacterial coexistence, we developed a mathematical model that describes the population dynamics of bacteria colonizing a bath of particles with a chosen dispersal strategy. More specifically, in our model, bacterial cells attach to particles from a free-living population in the bulk of the bath; they then grow and reproduce while attached. Detachment is stochastic with a fixed rate. After detachment, cells re-enter the free-living population and repeat the process. During the time spent attached to particles, all bacteria on a particle

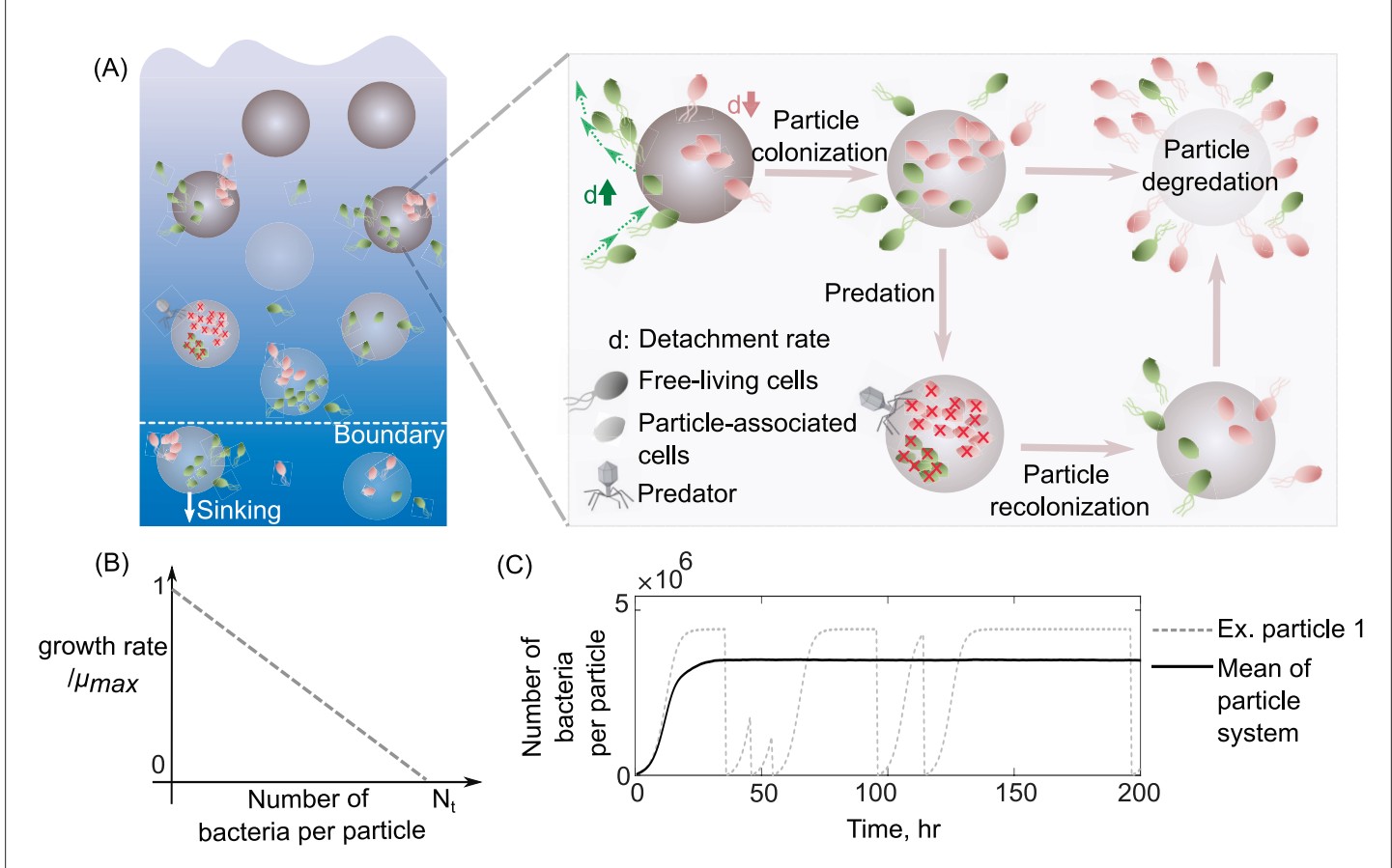

**Figure 1.** Schematic representation of the mathematical model simulating slow and fast dispersal strategies of bacterial populations that colonize particulate organic matter. (**A**) The model assumes that the predation on a particle or sinking out of system's boundaries kills its all associated populations. After infection, the noncolonized particle is then recolonized by free-living populations. As resources on a particle are consumed, its associated populations are dispersed and are added to the free-living populations. In this case, the old particle is replaced by a new uncolonized particle in the system. (**B**) The growth kinetics on a single particle is assumed to be density dependent and decreases linearly as a function of the number of cells colonizing the particle. $N_t$ represents the carrying capacity of the particle. (**C**) The dynamics of particle-associated cells and their corresponding growth rates are shown for a system with 1000 particles. The mean values over many particles and an example of dynamics on a single particle are illustrated.

The online version of this article includes the following figure supplement(s) for figure 1:

**Figure supplement 1.** Bacteria-particle encounter probability.

may die with a fixed probability per unit time, corresponding to their particle-wide mortality rate (*Figure 1A*). Another important feature of the model is density-dependent growth, which means that per capita growth rates decrease with increasing population size. For this, we use the classic logistic growth equation, which contains a simple linear density dependence (*Figure 1B*; Methods). Free-living subpopulations cannot grow, but die at a fixed mortality rate due to starvation. The probability of a bacterium encountering particles controls bacterial attachment, which we calculate using random walk theory as the hitting probability of two objects with defined sizes (*Leventhal et al., 2019*; *Frazier and Alber, 2012*; see Methods for details). We assume that the detachment rate is an intrinsic property of a bacterial population and comprises its dispersal strategy independent of the abiotic environment. In our simulations, it is the only trait that varies between different bacterial populations. Growing evidence has shown that bacterial detachment rates differ significantly across marine bacterial communities from solely planktonic cells to biofilm-forming cells on particles (*Yawata et al., 2014*; *Ebrahimi et al., 2019*). Using this mathematical model, we asked how variation in detachment rate affects bacterial growth dynamics and the ability of multiple subpopulations to coexist on particles.

For this, we simulated bacterial population dynamics on a bath of several particles and measured each population's relative abundance at a steady state (example in *Figure 1C*).

## Bacterial mortality determines optimal foraging strategies

Our model simulates growth, competition, and dispersal in a patchy landscape, similar to classical models of resource foraging, with the additional element of mortality, both within and outside patches (i.e., particles). We hypothesized that the inclusion of mortality could play an important role in affecting the success of a dispersal strategy (i.e., detachment rate), since it would alter the cost of staying on a particle. To investigate how mortality affects dispersal strategies, we studied its effect on the optimal strategy, which forms the focus of many classical models of foraging. According to optimal foraging theory (OFT), the optimal time spent on a particle is one that balances the time spent without food while searching for a new patch, with the diminishing returns from staying on a continuously depleting patch (*Yawata et al., 2020*; *Charnov, 1976*). In our model, particles are analogous to resource patches, and the detachment rate is simply the inverse of the time spent on a particle (residence time). We assumed that the optimal strategy maximizes the total biomass yield of the population.

As expected, OFT predicts the optimal detachment rate given a distribution of resources and search times, but only in the absence of mortality (*Figure 2A*). To test if our model agrees with the predictions of OFT, we calculated the optimal detachment rate ($d_{opt}$) using simulations of our model in the absence of mortality and compared it with OFT predictions (Methods). We found that the optimal detachment rate, which outcompetes all other detachment rates, was consistent with OFT predictions across a wide range of particle numbers in our system (*Figure 2A*). Strikingly, in the presence of mortality, the optimal detachment rate ($d_{opt}$) changed significantly, either increasing or decreasing depending on the type of mortality. When mortality was particle-wide, the optimal detachment rate was much higher than predicted by OFT, often resulting in residence times that were many days shorter than the OFT prediction (*Figure 2A*). This is because it is more beneficial to detach faster when there is a higher risk of particle-wide extinction. In contrast, when mortality was only present in free-living populations (affecting individuals, not particles, at a constant per capita rate), the optimal detachment rate was much lower than predicted by OFT (*Figure 2A*). These results expand on our knowledge of OFT and explain that the source and strength of mortality – on individuals or on whole particles – can differently impact the optimal detachment rate.

## A trade-off between growth and mortality enables the coexistence of dispersal strategies

Having observed that mortality can greatly affect the success of a dispersal strategy, we next sought to understand whether it could enable the coexistence of bacterial populations with different strategies (detachment rates). Simulations where we competed a pair of bacterial populations with different detachment rates revealed that differences in detachment rates alone are sufficient to enable coexistence on particles (*Figure 2B*). We assessed coexistence by measuring the relative abundances of populations at equilibrium (*Figure 2—figure supplement 1*). Interestingly, such a nontrivial coexistence only emerged in the presence of particle-wide mortality. In the absence of mortality on particles, we only observed trivial coexistence (coexisting populations had identical detachment rates, and for the purposes of the model, were one and the same; *Figure 2—figure supplement 2*). These results suggested that the presence of particle-wide mortality, where the entire population on a particle suffers rapid death, was crucial for populations with different dispersal strategies to coexist.

To investigate the underlying mechanisms that may give rise to the coexistence of populations with different detachment rates, we quantified the growth return of particle-associated populations as well as their survival rate on particles (*Figure 3A, B*). We calculated the average growth return based on the average number of offspring produced per capita during one single attachment–detachment event. The survival rate on particles was obtained by subtracting the mortality rate per capita from the offspring production rate per capita (*Figure 3B*; see Methods). The results revealed that a trade-off between bacterial growth return and survival rate emerged on particles, supporting the coexistence of populations with different detachment rates (*Figure 3C, D*). Populations that detach slowly from particles have higher growth returns but are also more susceptible to particle-associated mortality. In contrast, populations with low residence time on particles (high detachment rate) have low growth

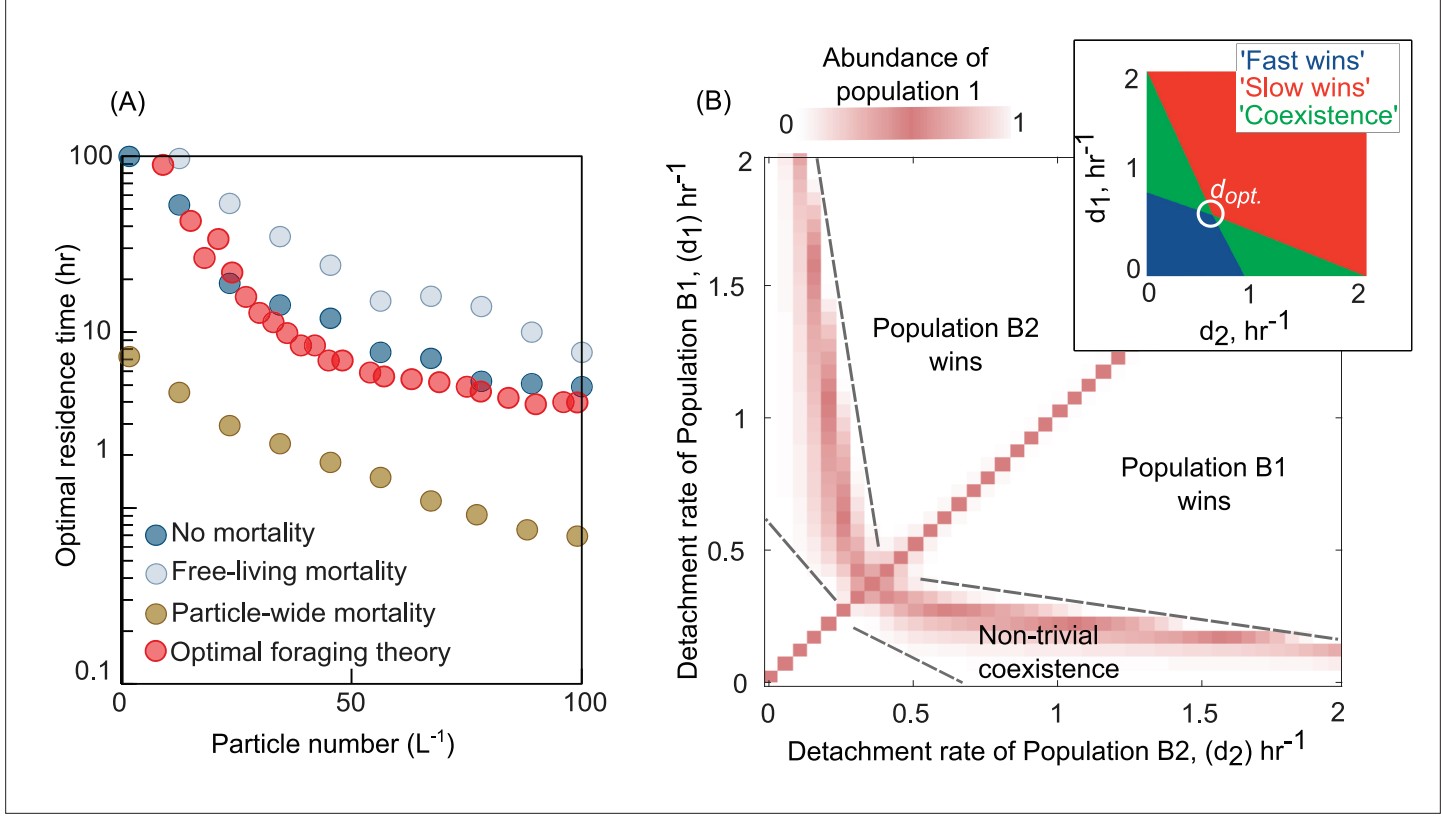

**Figure 2.** Variation in bacterial detachment strategies allow coexistence in the particle system. (**A**) Optimal residence time predicted by the population-based model and optimal foraging theory (OFT; Methods). Three scenarios with various particle-wide mortality ($m_p$) and mortality on free-living populations ($m_F$) are simulated with the following rates: (1) particle-wide mortality ($m_p = 0.05$ hr$^{-1}$, $m_F = 0.02$ hr$^{-1}$), (2) free-living mortality ($m_p = 0$ hr$^{-1}$, $m_F = 0.05$ hr$^{-1}$), and (3) no mortality ($m_p = 0$ hr$^{-1}$, $m_F = 0$ hr$^{-1}$). To calculate optimal residence time based on OFT, we used our model and tracked individual cells attaching to a particle. The time-averaged uptake rate of the attached cell and its instantaneous uptake rate were calculated. The residence time with similar instantaneous and time-averaged uptake rates is assumed to be optimal residence time based on OFT (see Method for details). In our population-based model, the optimal residence time is assumed to be a residence time that maximizes the growth return from the particles. (**B**) The relative abundance of population 1 is shown for competition experiments of two populations with different detachment rates. The relative abundance is measured at the equilibrium, where no changes in the sizes of both populations are observed. The area with white color represents the conditions where either one of the populations is extinct. The mortality on particles is assumed 0.02 hr$^{-1}$. (inset) Phase diagram of the coexistence as a function of detachment rates for two competing populations. $d_{opt}$ represents the optimal detachment rate that the coexistence range nears zero. (**B**) The attachment rates are kept constant at 0.0005 hr$^{-1}$. The number of particles is assumed to be 60 L$^{-1}$. The carrying capacity of the particle is assumed to be 5e10$^6$. Simulations are performed using our population-based mathematical model.

The online version of this article includes the following figure supplement(s) for figure 2:

**Figure supplement 1.** Two examples of population dynamics are shown wherein both populations reach a stable coexistence (I), while in the other scenario (II), one population is extinct.

**Figure supplement 2.** The relative abundance of population 1 is shown when no mortality on particles is considered for competition experiments of two populations with different detachment rates.

**Figure supplement 3.** In the absence of environmental fluctuations, competition experiment between populations with different detachment rates shows an emergence of an optimal detachment strategy that outcompete other populations.

**Figure supplement 4.** Cooperative growth kinetics restricts the coexistence range among two populations with different dispersal strategies.

**Figure supplement 5.** The sensitivity of coexistence among bacterial detachment strategies to competitive growth kinetic parameterizations (*Equation 6*: maximum growth rate $\mu_{max}$ and carrying capacity, $N_t$).

returns but they are less likely to die by predation or sink beyond the habitable zone. We next investigated whether such a trade-off was necessary to enable coexistence in our model.

We developed a coarse-grained model to address the conditions under which we might observe coexistence between populations whose only intrinsic difference was their detachment rates in our system. Our simple model expands on classical literature which describes coexistence among various

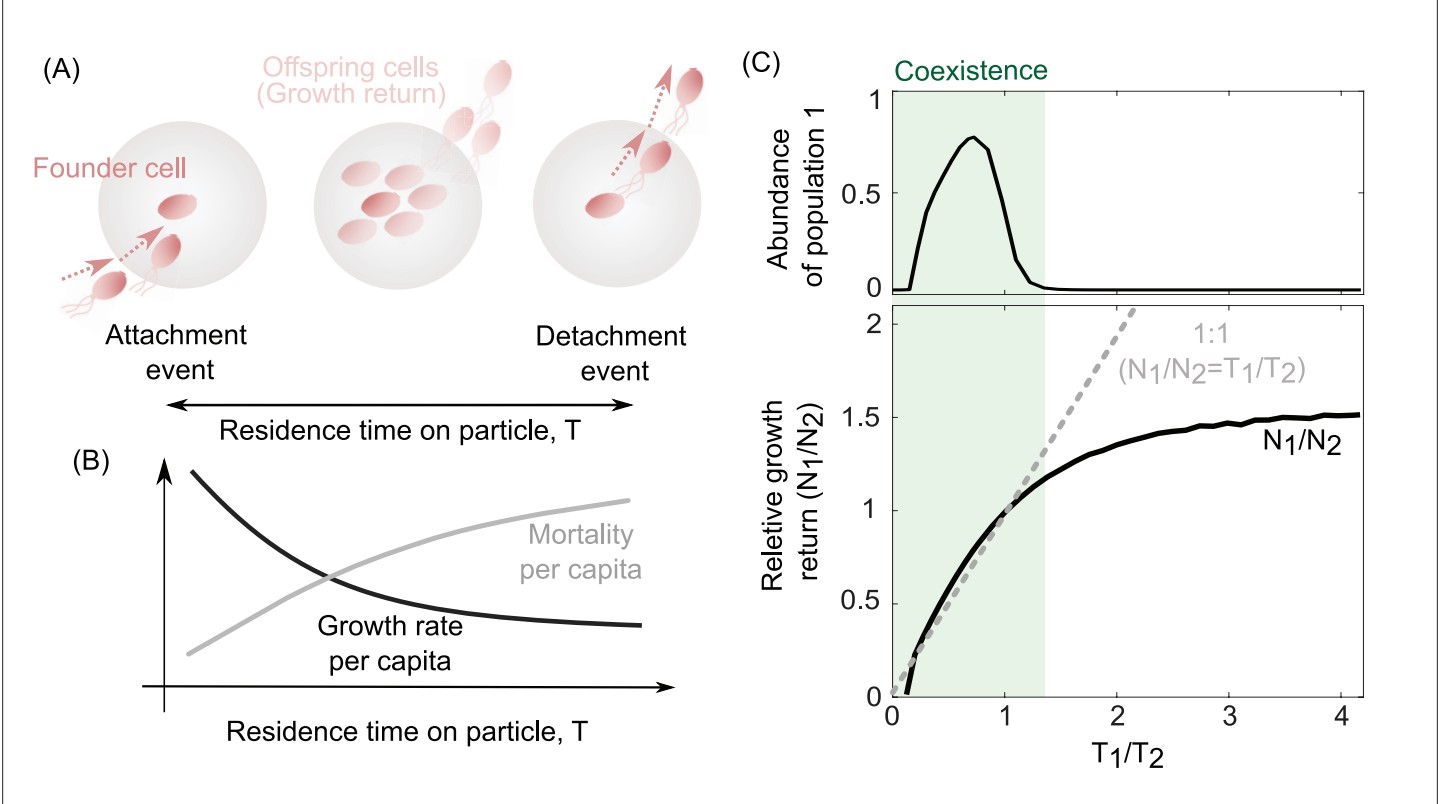

**Figure 3.** The trade-off between bacterial growth return and survival on the particles determines the coexistence range of competing populations. (**A**) The growth return of a single cell on a particle is calculated based on the number of offsprings produced during a single attachment/detachment event. (**B**) The growth rate on particles slows down as the particle is populated by offspring cells or new attaching cells that limit the net growth return from a particle. (**C**) The ratio of growth returns and survival of populations 1–2 per capita as a function of their radios of residence times is shown. The residence time on the particle is assumed to be the inverse of each population's detachment rate. The relative abundance of population 1 is shown for its corresponding simulations. The data are shown for the simulations where detachment rate of population 2 was kept constant at 0.2 hr$^{-1}$. Constant residence time for the population 2 (4 hr) is considered while varying the residence time of the first population across simulations. 1:1 line represents a coarse-grained model for the coexistence criteria of two competing populations.

dispersal strategies in spatially structured habitats (*Levin and Paine, 1974*; *Kneitel and Chase, 2004*; *Tilman, 1994*; *Amarasekare, 2003*). We simplified many details in favor of analytical tractability. Chiefly, we assumed that the growth dynamics on each particle were much faster than the dispersal dynamics across particles. This allowed us to replace detailed growth dynamics on single particles with a single number quantifying the bacterial population, $N$, after growth on each particle. In the model, we considered two particle-associated populations that competed for a shared pool of particles. To keep track of populations, we quantified the number of particles they had successfully colonized as $B_1$ and $B_2$, respectively. Individuals from both populations could detach from particles they had already colonized and migrate toa number $E$ of yet-unoccupied particles, with a rate proportional to their detachment rates, $d_1$ and $d_2$, respectively. Once migrated, individuals rapidly grew on unoccupied particles to their fixed per particle growth returns, $N_1$ and $N_2$. To model particle-wide mortality, we assumed a fixed per particle mortality rate, $m_p$. The population dynamics for the system of particles could therefore be written as follows:

$$\frac{dB_i}{dt} = (N_i d_i E - m_p)B_i \tag{1}$$

At equilibrium ($\frac{dB_i}{dt} = 0 \,\forall\, i$), either population can survive in the system if and only if its net colonization and mortality rates are equal ($N_i d_i E \approx m_p$). Consequently, the product of the growth return per particle and the detachment rate of either population should be equal ($N_1 d_1 \approx N_2 d_2$). By simplifying *Equation 1* at equilibrium, this model predicts that for two competing populations to coexist, their growth returns and detachment rates on particles must follow the relation:

$$\frac{N_1}{N_2} = \frac{d_2}{d_1} \qquad (2)$$

This relationship shows that coexistence demands a trade-off between the growth return ($N$) of a bacterial population, and its detachment rate ($d$), that is, the inverse of an individual's residence time on a particle. In other words, coexistence only emerges when the growth returns increase with the residence time on the particle ($\frac{N_1}{N_2} \sim \frac{T_1}{T_2}$). In agreement with this, simulations from our detailed model revealed that coexistence between two populations with different detachment rates only occurred in conditions where the two populations obeyed such a relationship, or trade-off (*Figure 3C*, gray region). We obtain the same relationship in *Equation 2* through an alternate calculation, where the relative abundances of both populations remain fixed, while the particle number varies.

While the trade-off in *Equation 2* allows coexistence and is necessary condition for it, it does not strictly hold across all parameter values, and hence prevents certain pairs of detachment rates to coexist (*Figure 3C*, white region). In particular, no detachment rate can coexist with the optimal detachment rate, thus rendering coexistence between any other set of detachment rates susceptible to invasion by this optimal strategy. Other strategies, when paired with the optimal strategy, disobey the condition in *Equation 2*, and thus cannot coexist with it. Therefore, if detachment rates were allowed to evolve, only one population would survive in the long run – the one with the optimal detachment rate (*Figure 2—figure supplement 3*). Motivated by this observation, we next asked whether environmental fluctuations would render coexistence evolutionarily stable, or whether they would further destabilize the coexistence of populations with nonoptimal dispersal strategies.

## Environmental fluctuations stabilize and enhance the diversity of dispersal strategies

The existence of a unique optimal strategy, even in the presence of particle-wide mortality (*Figure 2A*), suggests that the coexistence that we observed between populations with different detachment rates (*Figure 2B*) may not be evolutionarily stable. However, in the oceans, both the abundance of particles and the density of predators (such as phage) exhibit temporal and spatial fluctuations (*Nilsson et al., 2019*; *Garin-Fernandez et al., 2018*; *Luo et al., 2017*), in turn affecting the foraging dynamics of particle-associated bacterial populations. We used our model to study how the particle-wide mortality rate affects the likelihood of two particle-associated bacterial populations to coexist (see Methods). Surprisingly, we found a negative correlation between the mortality rate and particle abundance that enhances the range of coexistence among different detachment rates (*Figure 4A*). At low mortality rates, slow-detaching populations outcompete faster ones, as it is more advantageous to stay longer on particles and grow, that is, these populations derive higher net growth returns. However, a higher mortality rate on particles allows faster-detaching populations to instead gain an advantage over the slow-detaching populations, since they can better avoid particle-wide mortality events.

We extended our model to ask how variation in the total number of particles (or particle abundance) affect population dynamics and the coexistence range of populations with different dispersal strategies. The results indicated that an intermediate number of particles maximize the likelihood of coexistence of two populations with different dispersal strategies (*Figure 4A*). Here, we simulated a range of particle abundances, between 1 and 80 particles L$^{-1}$, which corresponds to the commonly observed range of particle abundances in aquatic environments (mean ~25 particles L$^{-1}$; *Figure 4—figure supplement 1*). Low particle abundances (0–20 L$^{-1}$) promote the growth of slow-detaching populations while at high particle abundances, fast-detaching populations dominate. The reason for this is the following: at particle abundances less than 20 L$^{-1}$, the probability of free-living cells finding and attaching to particles is less than 50% of the probability at high particle abundances (100 L$^{-1}$ in *Figure 1—figure supplement 1*). This makes particle search times very high, thus explaining how slow-detaching strategies have an advantage. As the number of particles increases, the entire system can support more cells (has a higher carrying capacity). This drives a decrease in particle search times, and thus increasingly advantages faster detaching strategies.

Interestingly, our results indicate that the optimal detachment rate ($d_{opt}$) is affected by the particle abundance and increases with the number of particles in the system (*Figure 4B*). We thus hypothesized that fluctuations in particle abundance may also induce fluctuations in the optimal detachment rate, such that no specific detachment rate would be uniquely favored at all times. Thus, environmental stochasticity would continuously change the optimal detachment rate; low particle

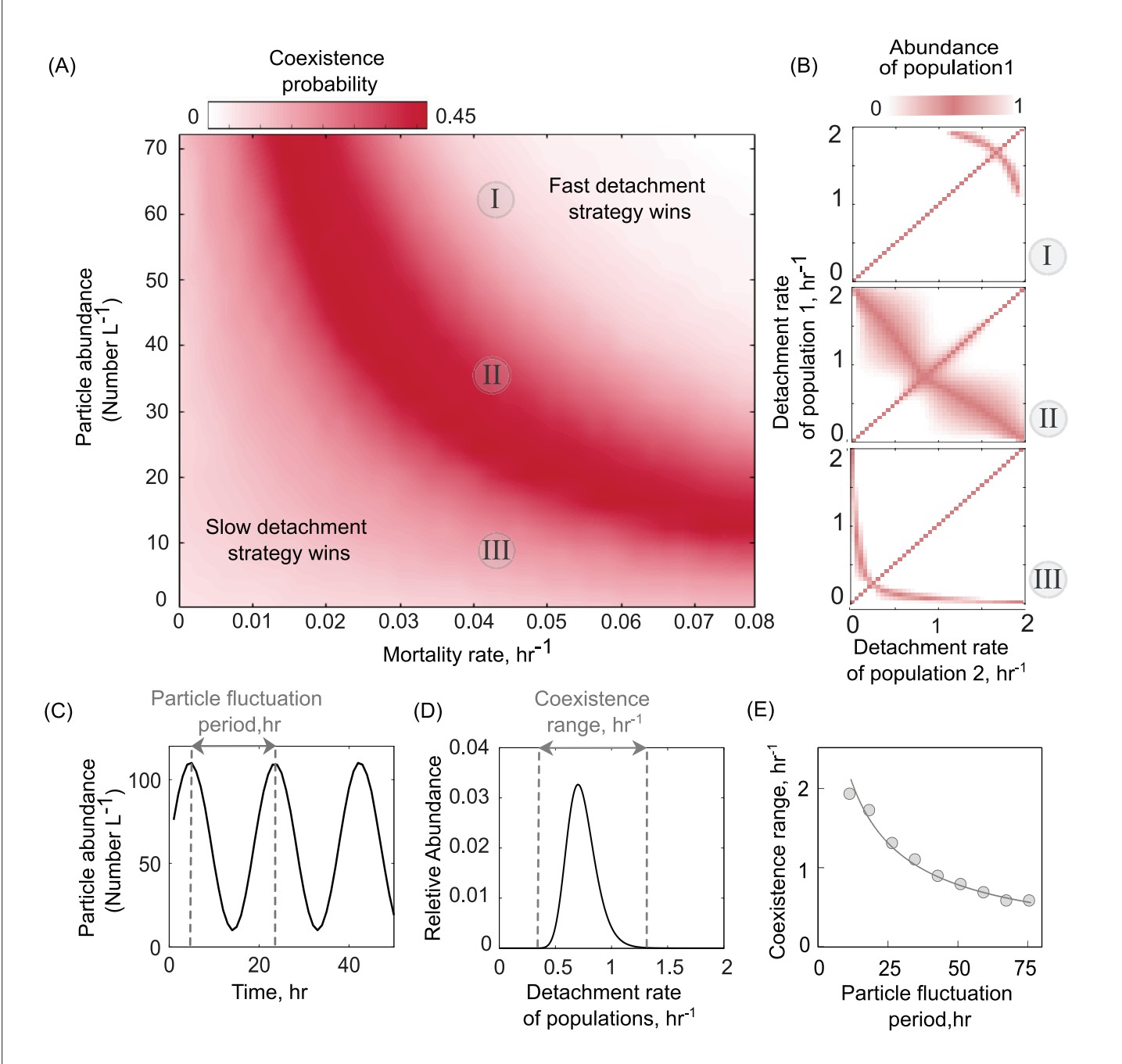

**Figure 4.** Particle abundance and predation rate shape the coexistence of populations with different dispersal strategies on the particle system. (**A**) The coexistence probability is shown for a range of particle abundances and predation rates. The coexistence probability is calculated by performing multiple competition experiments across populations with various detachment rates and quantifying the number of conditions that the coexistence between two populations is found. (**B**) For three particle abundances in **A**, the relative abundance of population 1 is shown in competition experiments of two populations. The numbers in circles refer to conditions in **A**. Simulations are assumed to be at the equilibrium when no changes in the size of either population are observed. The area with white color represents the conditions where either one of the populations is extinct. (**C**) A sine function is introduced to represent particle abundance fluctuations. (**D**) The coexistence range represents a range of detachment rates for populations that coexist at the equilibrium. Populations with relative abundances less than 5% of the most abundant population is assumed extinct. (**E**) The coexistence range is shown as a function of particle fluctuation period. The attachment rate and mortality rates are assumed to be ~0.0005 and ~0.045 hr$^{-1}$.

The online version of this article includes the following figure supplement(s) for figure 4:

**Figure supplement 1.** Particle abundance distributions extracted from the field observations.

**Figure supplement 2.** The durations of environmental fluctuation periods for particle abundances are extracted from field data *Lampitt et al., 1993*.

abundances would favor fast-detaching populations, while higher particle abundances would favor slow-detaching populations. Such a 'fluctuating optimum' may create temporal niches and promote higher bacterial diversity on marine particles. To test this hypothesis using our model, we simulated competition between 100 populations with different detachment rates under a periodically varying particle abundance (*Figure 4C*). The chosen frequencies of variation in particle abundance ($F_p$) were selected to be consistent with the observed frequencies in the ocean, with periods ranging between 10 and 100 hr (*Figure 4—figure supplement 2*; *Lampitt et al., 1993*). We quantified the range of detachment rates, a proxy for bacterial diversity, that could coexistat equilibrium (*Figure 4D*). The results revealed that the scenario with fastest fluctuations in particle numbers ($F_p = 10$ hr$^{-1}$) supported higher diversity among populations with different detachment rates (*Figure 4D*). Consistent with the fluctuation periods observed in the ocean, our simulations showed that fluctuation at the daily scale is sufficient to support the coexistence of different dispersal strategies. Overall, our model provides a framework to study how environmental fluctuations contribute to observed diversity in the dispersal strategies of particle-associated populations in marine environments.

## Discussion

In this study, we have shown a mechanism by which diverse dispersal strategies can coexist among bacterial populations that colonize and degrade POM in marine environments using a mathematical model. In our model, coexistence among populations with different dispersal strategies emerges from a trade-off between growth return and the probability of survival on particles. Such a trade-off determines the net number of detaching cells from particles that disperse into the bulk environment and colonize new particles. While slow-detaching populations are able to increase their growth return on particles and produce a relatively high number of offspring, they also experience higher mortality on particles that reduces their ability to colonize new particles. In contrast, faster-detaching populations can better avoid mortality by spending less time on particles, but this comes at the expense of lowering their growth return on a particle. Such populations can instead disperse and colonize a larger fraction of fresh yet-unoccupied particles. Interestingly, our results indicated that in the absence of mortality on particles, no coexistence is expected and there is a single dispersal strategy that provides the highest fitness advantage over dispersing populations, indicating that mortality on a particle is a key factor for the emergence of diverse dispersal strategies. Such correlated mortality with dispersal is the direct result of spatial structures created by particle-associated lifestyle, unlike the planktonic phase where predation probability per capita is expected to be uniform among planktonic cells. This study expands on the existing knowledge that spatial structure plays a critical role in promoting bacterial diversity in nature (*Eckburg et al., 2005*; *Zhang et al., 2014*), by incorporating the idea of particle-wide predation, which are events of correlated predation of an entire population on a particle. Such correlated predation could be an ecologically relevant mechanism that explains, in part, why we observe a higher diversity in particle-associated bacteria than planktonic bacteria in nature (*Milici et al., 2017*; *Ganesh et al., 2014*; *Crespo et al., 2013*). Our model assumes a general form of predation on particles that is insensitive to population type. However in the context of viral infection, field observations often show high strain specificity *Holmfeldt et al., 2007*; *Roux et al., 2012*; *Suttle and Chan, 1994*; *Suttle, 2005* that is likely to contribute to higher diversity in particle-associated populations. Viral infection act as a driving force to create a continuous succession of bacterial populations on particles by replacing phage exposed populations with less susceptible ones.

Consistent with the literature on OFT (*Fernandez et al., 2019*; *Taylor and Stocker, 2012*; *Vetter et al., 1998*), our model predicts the existence of an optimal foraging strategy for bacterial population colonizing particles in marine environments. Building on previous studies (e.g., *Yawata et al., 2020*) that show the optimal detachment rate is a function of search time for new resources, our study suggests that optimal detachment rate could be significantly impacted by the predation rate on particles. Our results indicated that a high mortality rate on particles shifts the optimal foraging strategy to populations with fast detachment rates. This finding agrees with previous OFT models that considered mortality, showing that optimal foraging effort and residence time on patches decrease significantly as the density of predators increase (*Newman, 1991*; *Abrams, 1993*). Interestingly, we showed that the variability in optimal detachment rate, due to environmental fluctuations in particle number and predation rate, could lead to evolutionarily stable coexistence among diverse dispersal strategies. Our results indicate that in the absence of any environmental fluctuations, there is a unique optimal

dispersal strategy. However, the optimal dispersal strategy depends on the abundance of particles, and thus fluctuations in their abundance at ecological timescales could sustain multiple dispersal strategies for long times. This finding is consistent with previous theoretical and empirical studies showing that environmental fluctuations such as light and temperature may lead to the stable coexistence of species (*Litchman, 2003*; *Li and Chesson, 2016*; *Catorci et al., 2017*; *Sousa, 1979*). Our model also predicts a loss of diversity when particle abundances significantly increase, consistent with field observations from algal blooms (*Teeling et al., 2012*; *West et al., 2008*; *Wemheuer et al., 2014*).

While we simplified bacterial colonization dynamics on particles by only considering competitive growth kinetics, variants of our model suggest that coexistence between different dispersal strategies is also expected under more complex microbial interactions observed on marine particles, including cooperative growth dynamics (*Figure 2—figure supplement 4*). Such simplifications allowed us to explore the role of dispersal in maintaining microbial diversity in natural systems, in addition to previously observed factors such as metabolic interaction, resource heterogeneities, and succession (*Muscarella et al., 2019*; *Datta et al., 2016*; *Dal Bello et al., 2021*). However, future studies, which can build on our model, could study how additional ecological factors contribute to bacterial marine diversity, such as complex trophic interactions leading to successional dynamics (*Boeuf et al., 2019*; *Datta et al., 2016*; *Lauro et al., 2009*; *Pascual-García et al., 2021*). Additionally, while we assumed diffusional searching for simplicity, extensions of our work could include more realistic bacterial search strategies, such as active motility and chemotaxis, which can play a big role in foraging in aquatic microorganisms (*Son et al., 2016*; *Stocker et al., 2008*). Finally, though we assumed a fixed detachment rate for each population, dispersal strategies can be quite complex, depending on local conditions such as bacterial and nutrient density on particles; a more thorough exploration of the relative costs and benefits of such myriad of dispersal strategies remains another promising avenue for future work. Overall, our model provides a reliable framework to further study how diverse dispersal strategies and mortality could contribute to the emergence of complex community dynamics on marine particles and how environmental factors impact microbial processes in regulating POM turnover at the ecosystem level.

## Materials and methods

In this study, a population-based model is developed that represents the interactions between the bacterial cells with different detachment rates and particles in a chemostat system, where the total number of particles is kept constant. The following provides a detailed procedure of the modeling steps as represented schematically in *Figure 1*. We have made the simulation code available in the following GitHub repository: https://github.com/alieb-mit-edu/Bacterial-dispersal-model.

### Modeling population dynamics on particles

Our model simulates the dynamics of two competing particle-associated populations ($B_p$) that colonize the same set of particles. Two populations ($i$ and $j$) are assumed to be identical, except for their detachment rates, $d$, from a particle ($d_i \neq d_j$). The dynamics of the particle-associated populations are determined by the rate at which cells attach to particles ($\alpha$) from the free-living population ($B_F$), the growth rate of attached cells ($\mu$) and detachment rate ($d$), as follows:

$$\frac{dB_{\mathrm{p},n,i}}{dt} = \alpha_i B_{\mathrm{F}i} + \mu_i \left( B_{\mathrm{p},n} \right) B_{\mathrm{p},n,i} - d_i B_{\mathrm{p},n,i} \tag{3}$$

where $n$ represents the particle index and its associated population, $i$. *Equation 3* can be formulated for any other population at the same particle. In a system with $N_p$ particles and $M$ populations, we numerically solve a finite set of equations ($N_p \times M$) at each time interval. The growth rate of population, $i$ ($\mu_i$ is a function of total particle-associated cells ($B_{p,n}$)), as described later in *Equation 6*.

From number conservation, the free-living bacterial pool $B_{Fi}$ of any population $i$ results from particle detachment and attachment dynamics. The rate of change of all free-living pools results from a combination of three factors: (1) the rate at which cells detach from the particles $d_i$, (2) the rate $\alpha_i$ at which cells attach to the particles, and (3) a mortality rate due to starvation $m_{Fi}$, as the following equation:

$$\frac{dB_{Fi}}{dt} = \sum_{n=1}^{N_p} d_i B_{p,n,i} - N_p \alpha_i B_{Fi} - m_{Fi} B_{Fi} \tag{4}$$

We run all dynamical simulations until an equilibrium is reached and there are no noticeable changes in the population size of particle-associated and free-living cells, that is, $\frac{dB_\text{P}}{dt} \approx 0$ and $\frac{dB_\text{F}}{dt} \approx 0$.

## Bacteria-particle encounter rate

We assume that a bacterial cell can attach to the particle it encounters and stay attached for a period of time ('residence time'). The encounter probability of a spherical cell with radius $r_c$ and a spherical particle with a radius of $r_p$ at a given time $t$ can be calculated using the hitting probability from random walk theory (*Leventhal et al., 2019*; *Frazier and Alber, 2012*):

$$P_e\left(i\right) = \frac{R}{r_{c,p}} erfc\left(\frac{r_{c,p} - R}{\sqrt{4D_t t}}\right) \tag{5}$$

where $R$ is the total radius ($R = r_p + r_c$), $D$ is an effective diffusion coefficient ($D = D_c + D_p$) for a bacterial cell (c) starting at a distance ($r_{c,p}$), and erfc represents the standard complementary error function (1 - erf). The diffusion coefficient can be calculated from an empirical model: $D = k_B T/6\pi r$, where $k_B \approx 1.38 \times 10^{-23}$ J K$^{-1}$ is Boltzmann's constant, $T = 293$ K is the ambient temperature, $\mu = 1.003$ mPa s is the viscosity of water at the given ambient temperature. In aquatic environments, the size of marine snow (>100 μm) is often a lot larger than the cell size, we thus assume that the effective diffusion is generally controlled by cell diffusion coefficient ($D \approx D_c$). From *Equation 5*, we calculate the total number attaching cells to a particle at a given time ($t$) from free-living cells of population $i$ by multiplying the hitting probability to the total number of free-living cells.

## Growth and reproduction on particles

We assume that per capita access to particulate resources decreases in proportion to the total number of cells that colonize the surface. This leads us to model bacterial competition on a given particle, $n$, with a linear negative density-dependent growth function.

In this model, we assume that the bacterial growth on the particle is competitive in which the growth rate, $\mu_i$ is not constant but changes as a function of the total biomass on a particle. The negative density-dependent growth is modeled by assuming a linear function with the total particle-associated cells ($B_{p,n} = \sum B_{p,n,i}$) on particle, $n$,

$$\begin{cases} \mu_i = \mu_\text{max}\left(1 - \frac{B_{p,n}}{N_\text{t}}\right) \\ \mu_i = 0, \quad B_{p,n} > N_\text{t} \end{cases} \tag{6}$$

where $B_{p,n} = \sum B_{p,n,i}$ represents the total number of particle-attached cells, $\mu_i$ represents that growth rate of population $i$, $\mu_{max}$ indicates the maximal growth rate, in the absence of competition, and $N_t$ represents the particle-specific carrying capacity. The net growth rate is assumed to be zero if more cells colonize a particle where bacteria have reached their carrying capacity; this occurs when bacteria have fully covered a particle's surface, such that the death or detachment of any cell is quickly replaced by the growth of another cell. The model assumes that free-living cells cannot grow. We performed a sensitivity analysis to competitive growth kinetic parameterizations (maximum growth rate $\mu_\text{max}$ and carrying capacity $N_t$) and showed that coexistence among bacterial detachment strategies is robust for a wide range of parameters (*Figure 2—figure supplement 5*).

Offspring production on the particle only occurs when particle-associated cells accumulate a total biomass that is larger than the biomass of a single cell ($m_d$). For simplicity, we only measured biomass based on the dry mass of the cells. The biomass accumulation rate on a particle for population $i$ is proportional to the available biomass on the particle, $n$ and its exponential growth rate ($\frac{dB_{p,n,i}}{dt} = B_{p,n,i}\mu_i$). With this, the total number of offspring ($N_{o,i}$) on a particle for a time interval of, $\Delta t$ can then be calculated as:

$$\begin{aligned} N_{\text{o},i} &= \frac{B_{p,n,i}\mu_i}{m_d}\Delta t, \quad B_{p,n,i}\mu_i\Delta t > m_d \\ N_{o,i} &= 0, \end{aligned} \tag{7}$$

## Particle-wide bacterial mortality

In the model, a general form of mortality on particles is considered that accounts for mortality induced by predation or particle sinking, taking cells beyond their preferred habitat. A constant fraction of particles ($m_p$) is randomly selected at each time interval ($\Delta t$) and their associated cells are removed from the particle. This fraction represents the particle-scale mortality rate ($m_p$). To maintain particle number equilibrium, a fraction $m_p$ of uncolonized particles is introduced into the system and colonized by free-living populations ($B_{p,n,i} = 0$).

Mortality of free-living cells is assumed to be caused by loss of biomass over a prolonged period of starvations from the absence of substrate uptake in the free-living phase. As described in *Equation 4*, free-living cells ($B_F$) lose a constant fraction of their biomass ($m_{Fi}$) every time step as the cell maintenance. Note that though detachment of cells from a particle appears similar to mortality on particles, in the former, detached cells move to the free-living pool, while in the latter, cells die and do not add to either pool.

## Particle degradation and turnover

We assume that a particle contains a finite amount of resources that is degraded by bacterial cells with a constant yield of converting the resources into biomass. From a previous study, we assume that the yield is about 5% and a significant fraction of particle degradation products are lost to the environment before being taken up by the cells (*Ebrahimi et al., 2019*).

## Optimal residence time from OFT

OFT describes the dispersal behavior of microbial populations in patchy environments assuming maximized growth return using the marginal value theorem. According to OFT, the growth return of particle-associated bacteria is maximized if a bacterial cell detaches from the particle when its time-averaged uptake rate reaches its instantaneous uptake rate. We applied this assumption to obtain the optimal residence time on particles by tracking individual cells in our model and numerically calculating their instantaneous uptake rate ($u(t)$) on a particle from the attachment time ($t_a$) to detachment using our population-based model. The residence time ($t_r$) is considered optimal when the following equation is satisfied (*Yawata et al., 2020*):

$$u\left(t_r\right) = \int_{t_a}^{t_r} \frac{u\left(t\right) dt}{\left(\tau_s + \left(t_r\left(\tau_s\right) - t_a\right)\right)} \tag{8}$$

where $\tau_s$ is the search time and a function of the number of particles in the system. We calculated the search time from *Equation 5* when the probability of the cell and particle encounter is above 95% (*Figure 1—figure supplement 1*).

## Acknowledgements

This work was supported by Simons Foundation: Principles of Microbial Ecosystems (PriME) award number 542,395. AE acknowledges funding from Swiss National Science Foundation: Grants P2EZP2 175,128 and P400PB_186751. AG acknowledges support from the Gordon and Betty Moore Foundation as a Physics of Living Systems Fellow through award number GBMF4513.

## Additional information

### Funding

| Funder | Grant reference number | Author |
| --- | --- | --- |
| Gordon and Betty Moore Foundation | GBMF4513 | Akshit Goyal |
| Simons Foundation | 542395 | Otto X Cordero |
| Swiss National Science Foundation | P2EZP2 175128 | Ali Ebrahimi |

| Funder | Grant reference number | Author |
|---|---|---|
| Swiss National Science Foundation | P400PB_186751 | Ali Ebrahimi |

The funders had no role in study design, data collection, and interpretation, or the decision to submit the work for publication.

## Author contributions

Ali Ebrahimi, Conceptualization, Methodology, Software, Validation, Visualization, Writing - original draft, Writing - review and editing; Akshit Goyal, Conceptualization, Formal analysis, Methodology, Visualization, Writing - original draft, Writing - review and editing; Otto X Cordero, Conceptualization, Investigation, Supervision, Writing - original draft, Writing - review and editing

## Author ORCIDs

Ali Ebrahimi ⓘ http://orcid.org/0000-0003-1079-7976
Akshit Goyal ⓘ http://orcid.org/0000-0002-9425-8269
Otto X Cordero ⓘ http://orcid.org/0000-0002-2695-270X

## Decision letter and Author response

Decision letter https://doi.org/10.7554/eLife.73948.sa1
Author response https://doi.org/10.7554/eLife.73948.sa2

---

# Additional files

## Supplementary files

• Transparent reporting form

## Data availability

Ours is a modeling and theoretical study, and has no associated data. All associated computer code relevant for the study and for reproducing the results is available as a GitHub repository at the following link: https://github.com/alieb-mit-edu/Bacterial-dispersal-model, (copy archived at swh:1:rev:9629ae0b5214a8a7a1ea9b96cef5d91adfe4a6ca).

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
