## [Editor Report]

This manuscript tackles an under-explored area in understanding microbial coexistence in marine and aquatic environments. This manuscript adds to the recently renewed interest on applications of optimal foraging theory to the study of microbial growth on marine snow.

---

## [Decision Letter]

**Decision letter after peer review:**

Thank you for submitting your article "Particle foraging strategies promote microbial diversity in marine environments" for consideration by *eLife*. Your article has been reviewed by 2 peer reviewers, and the evaluation has been overseen by a Reviewing Editor and Aleksandra Walczak as the Senior Editor. The following individual involved in review of your submission has agreed to reveal their identity: James O' Dwyer (Reviewer #1).

Essential revisions:

1) Ensure modeling choices are clearly explained and documented (including making code available)

2) Discuss the generality and limitations of the model and results

3) Provide additional background context/discussion of prior work as suggested by the reviewers

*Reviewer #1 (Recommendations for the authors):*

- In framing the paper, I think the authors are right to focus on dispersal and detachment as under-explored mechanisms. But readers will benefit from reference to other work (even on particle-associated microbes) related to resource diversity, succession, and crossfeeding. That can only help put the current study in context with other mechanisms for the maintenance of microbial diversity.

To expand on this, I know the authors have worked also on how microbial interactions, cross-feeding, and succession can maintain diversity, or at least add to our understanding of it on marine particles. Is the situation here envisioned significantly different than those experiments? If not, i think it is fine to focus on this different trait axis, and just consider the particles to in effect be a single resource, without resource preferences, etc. But I wanted to make sure I was understanding that correctly---maybe there is something different about the situation envisaged here that would make it less likely to have all of those other interactions (which clearly can contribute to many species being maintained). Martina Dal Ballo and Jeff Gore's work also seems relevant to this (where many resources are produced endogenously in an experimental system) and also work on resource heterogeneity in DOM in natural systems (e.g. Muscarella, Boot, Broeckling , Lennon).

Again, I don't think any of this detracts from the motivation for the present study. Just might fill out a fuller picture.

- There is a population growth process when a cell settles on a new particle. This is assumed to be logistic growth, though in the end, it seems likely that the precise dynamics of the growth process don't matter so much as the final abundance (carrying capacity). However, this seemed subtle to me for three reasons.

(i) It seems to me that the detachment rate should directly affect this final abundance---as an additional source of "mortality" (in the sense of removing individuals from the particle) contributing to the net growth rate. Maybe that effect will always be small, but this could be made clearer for readers if so.

(ii) The authors' conceptual diagram shows that one possible end point of this process is that the microbial population in effect eat the whole particle (Figure 1). This sharpens the issue to me of what the true dynamics are likely to be on a particle. For example, should I think of the population growing to a capacity that is roughly the surface area of the particle, and then gradually changing thereafter as the particle's physical size is reduced? And what direction would this change be? Could I think of a thickening of the film of particles and population size continuing to grow (maybe linearly) in time as interior layers continue to eat the particle? Or should I think of outer individuals as being shed, and the carrying capacity in effect reducing as the particle size reduces?

(iii) The issue of what is actually going to happen once a population reaches carrying capacity is also at the center of my final point here. It seems from the thought experiment above that it is unlikely that growth as such will stop when cells fill out the surface of a particle, since there are still resources to take up. So I am interested to know whether zero growth rate means to the authors that cells stop reproducing, or are cells dying to balance growth, or are they being shed from the particle?

It's possible that none of this matters too much if all that's important is a final population size. However, it might help to clarify the process for readers if we have a conceptual picture of what this final population size represents (surface of particle being filled? or volume of particle entirely eaten up) and if there is a truer picture of the dynamics than logistic growth.

- The relationship between the trade-off (between different detachment rates) derived in Equation 2 versus the optimal detachment rate (derived in the methods) is framed a little confusingly. If I understand correctly, the "trade-off" actually comes from the condition that a population will have net non-negative growth rate in the absence of other populations with different strategies. So it may be reasonable to frame this as a threshold---a necessary condition rather than a sufficient condition for a given population to persist. The reason I say this is that it is a bit confusing to have a trade-off that suggests a range of detachment rates can coexist so long as they differ in their carrying capacities, since it is then stated that the optimal detachment rate outcompetes all the others. Maybe I misunderstood something important being assumed about the carrying capacity for the optimal case, but a trade-off that also has an optimum is an odd outcome.

In short, it was not clear to me whether to populations satisfying this tradeoff in Equation (2) would tend to coexist. Or would in general one population (say the one closer to the optimum strategy) tend to outcompete the other? If so it might help to define more clearly what this tradeoff means. If I understood this correctly, I would not say that this issue merely indicates that the tradeoff is not "evolutionarily stable".

- In the end, it seems critical that for multiple strategies to be maintained in the population that there is not only whole-particle mortality (which in effect is highly correlated catastrophic dynamics for an individual microbial population), but that the inflow of resources itself fluctuates. Did I interpret that correctly? Readers may appreciate a slightly clearer description of how this environmental stochasticity differs from the previous possibility of whole-cell mortality, and this also left me wondering how to quantity the kind of environmental stochasticity that will generally lead to multiple strategies coexisting.

So my understanding from this was that whole-cell mortality was not on its own to avoid a single population outcompeting others. but I did not get such a clear picture of what environmental stochasticity WOULD allow for coexistence.

- One other reference that might be tangentially related, but I thought could be relevant: "The importance of being discrete: Life always wins on the surface" by Shnerb et al. This describes growth on particles in 2d or 3d, and shedding (which seems different from but not entirely different from detachment).

This could be an important reference because in this stochastic model, the effective growth rate is different from what you would have with the naive mean field model. So I am wondering if this might change any of the outcomes of Equations 1 and 2.

*Reviewer #2 (Recommendations for the authors):*

The authors investigated an interesting question related to the coexistence of bacterial species with different detachment strategies on particles. The results of this investigation are interesting and relevant for our understanding of microbial diversity in particle-associated communities, but the manuscript would benefit from a more in depth discussion of results from recent papers on microbial community dynamics of particles, many of which are cited in the current version but only in passing. There are also many possible extensions of this work, which the authors are probably aware of, which would be interesting to explore in future work. For example, the role of search strategies (e.g., random walks vs chemotaxis vs Levy walks) and detachment rates that depend non-linearly on the concentration of bacteria on the particle.

The authors seem to adopt a somewhat strict definition of Optimal Foraging Theory (OFP) limited to the Marginal Value Theorem (MVT). There are examples of OFT studies that do consider mortality and predation risk, finding that predictions of the MVT do not hold in these settings, e.g.:

Abrams, PA. "Optimal traits when there are several costs: the interaction of mortality and energy costs in determining foraging behavior." Behavioral Ecology, vol. 4, no. 3, 1993, pp. 246-259.

Newman, JA. " Patch use under predation hazard: foraging behavior in a simple stochastic environment." Oikos, vol. 61, 1991,

pp. 29-44.

It would be important to know how this study relates to previous results of OFT that do include mortality and predation risk.

In its current form, there are a few places where better description of the numerical simulations performed would critically enhance the manuscript and the reproducibility of the results. Specifically:

- There is a deterministic particle mortality rate m_p,i_ in Equation 3, and an additional, stochastic particle mortality in the section "Bacterial mortality". Are the two implemented with the same rate, and what is the rationale for implementing both forms of mortality? The manuscript text only seems to describe the stochastic particle mortality, but is never too explicit about it. As described in lines 385-394, it seems that the stochastic mortality rate depends on the numerical integration time step: because m_p_ is a rate, a fraction m_p_ * dt of particles, where dt is the integration time step, should be chosen at each time step to impose mortality, rather than a "fraction" m_p_ as suggested by the text at lines 387-389.

- The most problematic section is "Bacteria-particle encounter rate". The authors mention explicitly the encounter probability of spherical cells undergoing random walks, but they need a rate to incorporate in the equations via parameter α. An explicit expression for α is not provided, and I would have expected α to be the diffusive flux towards a spherical absorber (see, e.g., Berg's "Random walks in biology" page 27, Equation 2.20): I = 4 \pi D R C_0_, where C_0_ is the concentration of detached bacteria, but this is not mentioned. Also, what is d in Equation 5? It should be D_c,p_, and it can't be the detachment rate d. The estimate for the diffusion coefficient of cells via the Einstein-Stokes relationship is too small if bacteria are motile, as suggested by Figure 1. For motile cells, cell diffusion can be orders of magnitude larger than the Einstein-Stokes estimate (see, e.g., Berg's "Random walks in biology" page 93 – Movement of self-propelled objects). The sentence "From Equation 5, we calculated the total number attaching cells to a particle at a given time (t) from free living cells of population i by multiplying the hitting probability to the total number of free-living cells" is very hard for me to interpret: which choice of D_c,p_ was used in Equation 5? I would also mention explicitly that this entire section assumes instantaneous attachment of bacteria to particles with an infinite rate coefficient.

For a study that is mostly numerical such as this one, availability of the computer code for peer review would greatly enhance the reproducibility of the results and would have clarified some of the doubts expressed above. I would encourage the authors to post it on Github for peer review, or provide it as supplementary material with the submission.

It would be very informative to know under which conditions the analytical approximation described at lines 174-216 breaks down. At low particle density, the search time may be much longer than the growth period on the particle, but at high particle densities this may not be true. Would the approximation work less well in those conditions?

---

## [Author Response]

Essential revisions:1) Ensure modeling choices are clearly explained and documented (including making code available)

Following the suggestions of all reviewers, the revised manuscript contains new text detailing our model choices and explaining the rationale behind them. We have also made all the simulation code available in the following GitHub repository: https://github.com/alieb-mit-edu/Bacterial-dispersal-model

2) Discuss the generality and limitations of the model and results

In the revised manuscript, we now discuss our interpretation of the model, including how its assumptions can be generalized; we have also added text discussing its limitations.

3) Provide additional background context/discussion of prior work as suggested by the reviewers

We are grateful to the reviewers for offering suggestions on appropriately crediting prior work; the revised manuscript now contains new citations and references, which discuss how our work connects with past literature on e.g., optimal foraging theory.

Reviewer #1 (Recommendations for the authors):- In framing the paper, I think the authors are right to focus on dispersal and detachment as under-explored mechanisms. But readers will benefit from reference to other work (even on particle-associated microbes) related to resource diversity, succession, and crossfeeding. That can only help put the current study in context with other mechanisms for the maintenance of microbial diversity.To expand on this, I know the authors have worked also on how microbial interactions, cross-feeding, and succession can maintain diversity, or at least add to our understanding of it on marine particles. Is the situation here envisioned significantly different than those experiments? If not, i think it is fine to focus on this different trait axis, and just consider the particles to in effect be a single resource, without resource preferences, etc. But I wanted to make sure I was understanding that correctly---maybe there is something different about the situation envisaged here that would make it less likely to have all of those other interactions (which clearly can contribute to many species being maintained). Martina Dal Ballo and Jeff Gore's work also seems relevant to this (where many resources are produced endogenously in an experimental system) and also work on resource heterogeneity in DOM in natural systems (e.g. Muscarella, Boot, Broeckling , Lennon).Again, I don't think any of this detracts from the motivation for the present study. Just might fill out a fuller picture.

Thank you for the constructive comment. We now expanded our discussion to include this point in the revised manuscript (lines 322-336):

“While we simplified bacterial colonization dynamics on particles by only considering competitive growth kinetics, variants of our model suggest that coexistence between different dispersal strategies is also expected under more complex microbial interactions that are observed on marine particles, including cooperative growth dynamics (Figure 1 —figure supplement 1). Such simplifications allowed us to explore the role of dispersal in maintaining microbial diversity in natural systems, in addition to previously observed factors such as metabolic interaction, resource heterogeneities and succession (Datta et al. 2016; Dal Bello et al. 2021; Muscarella et al. 2019). Recent studies have observed the emergence of complex trophic interactions and successional dynamics across particle-associated communities that our model can provide a theoretical framework to evaluate contributions on such mechanisms on maintaining bacterial diversity (Lauro et al. 2009; Datta et al. 2016; Pascual-García et al. 2021; Boeuf et al. 2019).”

- There is a population growth process when a cell settles on a new particle. This is assumed to be logistic growth, though in the end, it seems likely that the precise dynamics of the growth process don't matter so much as the final abundance (carrying capacity). However, this seemed subtle to me for three reasons.(i) It seems to me that the detachment rate should directly affect this final abundance---as an additional source of "mortality" (in the sense of removing individuals from the particle) contributing to the net growth rate. Maybe that effect will always be small, but this could be made clearer for readers if so.

Yes, the reviewer is correct. In the revised manuscript, we have clarified the distinction between particle-wide mortality and detachment in how affect bacterial numbers both on particles and in the total system (lines 424-426).

The detachment rate can be seen as a source of mortality if the mortality is defined as removing cells from the particle. It should be noted that the cells detaching from the particle will be added to the free-living population (BF) which can have a positive effect on the rate of attachment to the particle (a*BF). Therefore, the "mortality" induced by the detachment rate is different than the particle-wide mortality (mp) which removes cells from the system and reduces the total number of free-living and particle-associated cells (BF + BP). Since we performed our simulations for a wide range of detachment rates, the rate of cells removed by this effect could be comparable to or higher than the particle-wide mortality in a small fraction of the simulations, but not large overall.

“Note that though detachment of cells from a particle appears similar to mortality on particles, in the former, detached cells move to the free-living pool, while in the latter, cells die and do not add to either pool.”

(ii) The authors' conceptual diagram shows that one possible end point of this process is that the microbial population in effect eat the whole particle (Figure 1). This sharpens the issue to me of what the true dynamics are likely to be on a particle. For example, should I think of the population growing to a capacity that is roughly the surface area of the particle, and then gradually changing thereafter as the particle's physical size is reduced? And what direction would this change be? Could I think of a thickening of the film of particles and population size continuing to grow (maybe linearly) in time as interior layers continue to eat the particle? Or should I think of outer individuals as being shed, and the carrying capacity in effect reducing as the particle size reduces?

As the reviewer understands, we do not explicitly model the size and food content on a particle, only an implicit density-dependent net growth rate of a bacterial population on the particle. Nevertheless, we believe our model captures the process of a population growing on a particle’s surface and eventually covering its entirety, rather than a volumetric shrinking over time as the particle is consumed. This is because the typical time taken to completely consume a particle is much larger than the time it takes for the particle to sink below a certain ocean depth and lost to bacterial grazing.

In the revised manuscript, we have clarified our interpretation of bacterial growth in the model (lines 388-395).

(iii) The issue of what is actually going to happen once a population reaches carrying capacity is also at the center of my final point here. It seems from the thought experiment above that it is unlikely that growth as such will stop when cells fill out the surface of a particle, since there are still resources to take up. So I am interested to know whether zero growth rate means to the authors that cells stop reproducing, or are cells dying to balance growth, or are they being shed from the particle?

In our model, a zero net growth rate of the population on a particle implies that cellular reproduction balances intrinsic cell death (not extrinsic whole-particle mortality) and detachment. Thus, the number of cells on the particles reaches a steady state.

It's possible that none of this matters too much if all that's important is a final population size. However, it might help to clarify the process for readers if we have a conceptual picture of what this final population size represents (surface of particle being filled? or volume of particle entirely eaten up) and if there is a truer picture of the dynamics than logistic growth.

We thank the reviewer for detailing this concern. In the revised manuscript, we have added clarifying text which explicitly explains our interpretation of the population size on each individual particle in the model (lines 388-395). Specifically, as a particle is colonized, the bacterial population on it grows in number and density, eventually covering the surface of the particle. At carrying capacity, there is no more space to grow – the population size reaches a steady state – and any spontaneously dying cells are replaced by newly growing ones.

“We assume that bacteria grow on each particle, covering its surface over time; as the surface gets covered, bacterial density increases, in turn slowing down growth. The net growth rate is assumed to be zero if more cells are colonizing the particle compared to its carrying capacity; this occurs when bacteria have fully covered a particle’s surface, such that the death or detachment of any cell is quickly replaced by the growth of another cell.”

- The relationship between the trade-off (between different detachment rates) derived in Equation 2 versus the optimal detachment rate (derived in the methods) is framed a little confusingly. If I understand correctly, the "trade-off" actually comes from the condition that a population will have net non-negative growth rate in the absence of other populations with different strategies. So it may be reasonable to frame this as a threshold---a necessary condition rather than a sufficient condition for a given population to persist. The reason I say this is that it is a bit confusing to have a trade-off that suggests a range of detachment rates can coexist so long as they differ in their carrying capacities, since it is then stated that the optimal detachment rate outcompetes all the others. Maybe I misunderstood something important being assumed about the carrying capacity for the optimal case, but a trade-off that also has an optimum is an odd outcome.In short, it was not clear to me whether to populations satisfying this tradeoff in equation (2) would tend to coexist. Or would in general one population (say the one closer to the optimum strategy) tend to outcompete the other? If so it might help to define more clearly what this tradeoff means. If I understood this correctly, I would not say that this issue merely indicates that the tradeoff is not "evolutionarily stable".

We understand the source of confusion and are happy to clarify. In the revised manuscript, we have also clarified the text and explained the tradeoff more carefully.

For all populations, there is a trade-off between detachment rate and growth return (population size per particle at steady state) due to the dynamics of the model; as one rises, the other falls. However, the mathematical relationship between the two quantities is complicated and parameter-dependent; in some parameter region, growth return changes rapidly with detachment rate, while in another, it changes relatively slowly.

For two populations to coexist, their detachment rates must obey equation (2). Equation (2) is a necessary condition and demands a specific mathematical form of the tradeoff between detachment rate and growth return. As Figure 3C shows, not all pairs of populations satisfy this equation; those that do can indeed coexist. Typically, the growth return is a complex quantity that depends on various parameter choices regarding the on-particle dynamics, and cannot be obtained using our coarse-grained description.

When one of the two populations has a detachment rate equal to the optimal detachment rate, the second population always fails to satisfy equation (2), i.e., the ratio of growth returns of the pair lies in the white region of Figure 3C. Thus, no population can coexist with the optimal detachment rate. Lines 211-217:

“While the trade-off in Equation 2 allows coexistence and is necessary condition for it, it does not hold across all parameter values, and does not allow any pair of detachment rates to coexist (Figure 3C, white region). In particular, no detachment rate can coexist with the optimal detachment rate, thus rendering coexistence between any other set of detachment rates susceptible to invasion by this optimal strategy. Other strategies, when paired with the optimal strategy, disobey the condition in Equation 2, and thus cannot coexist with it.”

- In the end, it seems critical that for multiple strategies to be maintained in the population that there is not only whole-particle mortality (which in effect is highly correlated catastrophic dynamics for an individual microbial population), but that the inflow of resources itself fluctuates. Did I interpret that correctly? Readers may appreciate a slightly clearer description of how this environmental stochasticity differs from the previous possibility of whole-cell mortality, and this also left me wondering how to quantity the kind of environmental stochasticity that will generally lead to multiple strategies coexisting.

This is a good point. In the revised text, we have clarified the difference between whole-particle mortality and environmental stochasticity, which brings in new particles. In general, any event that changes the number of empty particles should aid in the coexistence of multiple strategies. Particles can be emptied by high levels of whole-particle mortality, or new empty particles can be brought in by events such as algal blooms (environmental stochasticity).

So my understanding from this was that whole-cell mortality was not on its own to avoid a single population outcompeting others. but I did not get such a clear picture of what environmental stochasticity WOULD allow for coexistence.

We have now added new text providing intuition for how environmental stochasticity promotes coexistence despite an optimal detachment rate. The optimal detachment rate is not fixed, but depends on the total number of particles and the whole-particle mortality rate. What environmental stochasticity, i.e., changing the total number of particles, does is that it changes the optimal detachment rate over time. Thus, two populations (fast and slow detaching) can coexist if the environment keeps oscillating between regimes where either population is advantaged. Lines: 254-262:

“We thus hypothesized that fluctuations in particle abundance may also induce fluctuations in the optimal detachment rate, such that no specific detachment rate would be uniquely favored at all times. Thus, environmental stochasticity would constantly change the optimal detachment rate; low particle abundances would favor fast-detaching populations, while higher particle abundances would favor slow-detaching populations. Such a “fluctuating optimum” may create temporal niches and promote higher bacterial diversity on marine particles.”

- One other reference that might be tangentially related, but i thought could be relevant: "The importance of being discrete: Life always wins on the surface" by Shnerb et al. This describes growth on particles in 2d or 3d, and shedding (which seems different from but not entirely different from detachment).This could be an important reference because in this stochastic model, the effective growth rate is different from what you would have with the naive mean field model. So I am wondering if this might change any of the outcomes of Equations 1 and 2.

Thank you very much for sharing this important reference with us. This is a valid point. The spatial distribution of cells on the particle could lead to a drastically different form of growth that may not be captured by mean-field models. In fact, in our previous study, we have shown that bacterial cells can spatially self-organize on the particles allowing them to cooperate and grow on recalcitrate particles. And without spatial organization, cells would go extinct.

We have performed sensitivity analyses for various growth dynamics on the particle (cooperative vs competitive) in Figure 2 —figure supplement 4. We have shown that our observations of the emergence of coexistence is robust to the form of growth on particles.

More studies are needed to explore the role that spatial distribution of cells could play in shaping the coexistence of cells on particles.

Reviewer #2 (Recommendations for the authors):The authors investigated an interesting question related to the coexistence of bacterial species with different detachment strategies on particles. The results of this investigation are interesting and relevant for our understanding of microbial diversity in particle-associated communities, but the manuscript would benefit from a more in depth discussion of results from recent papers on microbial community dynamics of particles, many of which are cited in the current version but only in passing. There are also many possible extensions of this work, which the authors are probably aware of, which would be interesting to explore in future work. For example, the role of search strategies (e.g., random walks vs chemotaxis vs Levy walks) and detachment rates that depend non-linearly on the concentration of bacteria on the particle.

Thanks for the constructive comments. In the revised manuscript, we have expanded our discussion to include assumptions of the model and opportunities for future works in the discussion (lines 322-336):

“While we simplified bacterial colonization dynamics on particles by only considering competitive growth kinetics, variants of our model suggest that coexistence between different dispersal strategies is also expected under more complex microbial interactions that are observed on marine particles, including cooperative growth dynamics (Figure 2 —figure supplement 4). Such simplifications allowed us to explore the role of dispersal in maintaining microbial diversity in natural systems, in addition to previously observed factors such as metabolic interaction, resource heterogeneities and succession^8,52,53^. Recent studies have observed the emergence of complex trophic interactions and successional dynamics across particle-associated communities that our model can provide a theoretical framework to evaluate contributions on such mechanisms on maintaining bacterial diversity ^18,52,54,55^. In addition, the current work only assumes Brownian motion while bacterial active motility and chemotaxis are shown to play a big role in the foraging strategy of aquatic microorganisms^56–58^. Overall, our model provides a reliable framework to further study how diverse dispersal strategies and mortality could contribute to the emergence of complex community dynamics on marine particles and how environmental factors impact microbial processes in regulating POM turnover at the ecosystem level. ”

The authors seem to adopt a somewhat strict definition of Optimal Foraging Theory (OFP) limited to the Marginal Value Theorem (MVT). There are examples of OFT studies that do consider mortality and predation risk, finding that predictions of the MVT do not hold in these settings, e.g.:Abrams, PA. "Optimal traits when there are several costs: the interaction of mortality and energy costs in determining foraging behavior." Behavioral Ecology, vol. 4, no. 3, 1993, pp. 246-259.Newman, JA. " Patch use under predation hazard: foraging behavior in a simple stochastic environment." Oikos, vol. 61, 1991,pp. 29-44.It would be important to know how this study relates to previous results of OFT that do include mortality and predation risk.

Thank you very much for sharing these important articles. We carefully read those articles that include mortality in the formulation of OFT. Interestingly, our work is consistent with the previous publications in showing that the presence of mortality/predation reduces residence time on the particle/patch.

We have now expanded our discussion to make this comparison (lines 309-311).

“This finding agrees with previous OFT models that considered mortality, showing that optimal foraging effort and residence time on patches decrease significantly as the density of predators increase (Newman 1991; Abrams 1993).”

In its current form, there are a few places where better description of the numerical simulations performed would critically enhance the manuscript and the reproducibility of the results. Specifically:- There is a deterministic particle mortality rate m_p,i_ in Equation 3, and an additional, stochastic particle mortality in the section "Bacterial mortality". Are the two implemented with the same rate, and what is the rationale for implementing both forms of mortality? The manuscript text only seems to describe the stochastic particle mortality, but is never too explicit about it. As described in lines 385-394, it seems that the stochastic mortality rate depends on the numerical integration time step: because m_p_ is a rate, a fraction m_p_ * dt of particles, where dt is the integration time step, should be chosen at each time step to impose mortality, rather than a "fraction" m_p_ as suggested by the text at lines 387-389.

Thanks for pointing out this. This is correct and we now modified the equation or the text to reflect the correct way of implementing the mortality on the particle. We removed mortality term in Equation 3 which was creating confusion. There is only one mortality that is considered and that is the one described in the second of “Bacterial mortality”. We now included time step, Δt and particle number in this section to reflect how the fraction of particles are determined for mortality (Lines: 411-417).

“In the model, a general form of mortality on particles is considered that accounts for mortality induced by predation or particle sinking and becoming inaccessible at deeper layers of the water column in the ocean. A constant fraction of particles (N_p_.m_p_.Δt) that represent the mortality rate at particle scale (m_p_) is uniform randomly selected at each time interval (Δt) and their associated cells are removed from the particle. Then, the uncolonized particle is again introduced into the system and colonized by free-living populations (Bp,n,i=0). ”

- The most problematic section is "Bacteria-particle encounter rate". The authors mention explicitly the encounter probability of spherical cells undergoing random walks, but they need a rate to incorporate in the equations via parameter α. An explicit expression for α is not provided, and I would have expected α to be the diffusive flux towards a spherical absorber (see, e.g., Berg's "Random walks in biology" page 27, Equation 2.20): I = 4 \pi D R C_0_, where C_0_ is the concentration of detached bacteria, but this is not mentioned. Also, what is d in Equation 5? It should be D_c,p_, and it can't be the detachment rate d. The estimate for the diffusion coefficient of cells via the Einstein-Stokes relationship is too small if bacteria are motile, as suggested by Figure 1. For motile cells, cell diffusion can be orders of magnitude larger than the Einstein-Stokes estimate (see, e.g., Berg's "Random walks in biology" page 93 – Movement of self-propelled objects). The sentence "From Equation 5, we calculated the total number attaching cells to a particle at a given time (t) from free living cells of population i by multiplying the hitting probability to the total number of free-living cells" is very hard for me to interpret: which choice of D_c,p_ was used in Equation 5? I would also mention explicitly that this entire section assumes instantaneous attachment of bacteria to particles with an infinite rate coefficient.For a study that is mostly numerical such as this one, availability of the computer code for peer review would greatly enhance the reproducibility of the results and would have clarified some of the doubts expressed above. I would encourage the authors to post it on Github for peer review, or provide it as supplementary material with the submission.

The code is now available online at the following URL: https://github.com/alieb-mit-edu/Bacterial-dispersal-model

It would be very informative to know under which conditions the analytical approximation described at lines 174-216 breaks down. At low particle density, the search time may be much longer than the growth period on the particle, but at high particle densities this may not be true. Would the approximation work less well in those conditions?

The reviewer is correct: as the density of particles increases, the time taken for bacteria to colonize new particles will decrease. Certainly, when particle densities increase to be very large (by more than an order of magnitude), our approximation will break down. However, for the numbers in our study, where particle densities vary by 2-3x, our numerics support our assumption that the approximation works reasonably well.

References:

Abrams, Peter A. 1993. “Optimal Traits When There Are Several Costs: The Interaction of Mortality and Energy Costs in Determining Foraging Behavior.” *Behavioral Ecology* 4 (3): 246–59. https://doi.org/10.1093/beheco/4.3.246.

Boeuf, Dominique, Bethanie R. Edwards, John M. Eppley, Sarah K. Hu, Kirsten E. Poff, Anna E. Romano, David A. Caron, David M. Karl, and Edward F. DeLong. 2019. “Biological Composition and Microbial Dynamics of Sinking Particulate Organic Matter at Abyssal Depths in the Oligotrophic Open Ocean.” Proceedings of the National Academy of Sciences of the United States of America. https://doi.org/10.1073/pnas.1903080116.

Dal Bello, Martina, Hyunseok Lee, Akshit Goyal, and Jeff Gore. 2021. “Resource–Diversity Relationships in Bacterial Communities Reflect the Network Structure of Microbial Metabolism.” *Nature Ecology & Evolution* 5 (10): 1424–34. https://doi.org/10.1038/s41559-021-01535-8.

Datta, Manoshi S, Elzbieta Sliwerska, Jeff Gore, Martin Polz, and Otto X Cordero. 2016. “Microbial Interactions Lead to Rapid Micro-Scale Successions on Model Marine Particles.” *Nature Communications* 7: 11965. https://doi.org/10.1038/ncomms11965.

Lauro, Federico M, Diane McDougald, Torsten Thomas, Timothy J Williams, Suhelen Egan, Scott Rice, Matthew Z DeMaere, et al. 2009. “The Genomic Basis of Trophic Strategy in Marine Bacteria.” *Proceedings of the National Academy of Sciences* 106 (37): 15527 LP – 15533. https://doi.org/10.1073/pnas.0903507106.

Muscarella, Mario E., Claudia M. Boot, Corey D. Broeckling, and Jay T. Lennon. 2019. “Resource Heterogeneity Structures Aquatic Bacterial Communities.” *ISME Journal*. https://doi.org/10.1038/s41396-019-0427-7.

Newman, Jonathan A. 1991. “Patch Use under Predation Hazard: Foraging Behavior in a Simple Stochastic Environment.” *Oikos* 61 (1): 29–44. https://doi.org/10.2307/3545404.

Pascual-García, Alberto, Julia Schwartzman, Tim Enke, Arion Iffland-Stettner, Otto Cordero, and Sebastian Bonhoeffer. 2021. “Turnover in Life-Strategies Recapitulates Marine Microbial Succession Colonizing Model Particles.” bioRxiv. https://doi.org/10.1101/2021.11.05.466518.